# Empowering Clinicians with Medical Decision Transformers: A Framework for Sepsis Treatment

## Abstract

Offline reinforcement learning has shown promise for solving tasks in safety-critical settings, such as clinical decision support. Its application, however, has been limited by the lack of interpretability and interactivity for clinicians. To address these challenges, we propose the *medical decision transformer (MeDT)*, a novel and versatile framework based on the goal-conditioned reinforcement learning paradigm for sepsis treatment recommendation. *MeDT* uses the decision transformer architecture to learn a policy for drug dosage recommendation. During offline training, *MeDT* utilizes collected treatment trajectories to predict administered treatments for each time step, incorporating known treatment outcomes, target acuity scores, past treatment decisions, and current and past medical states. This analysis enables *MeDT* to capture complex dependencies among a patient's medical history, treatment decisions, outcomes, and short-term effects on stability. Our proposed conditioning uses acuity scores to address sparse reward issues and to facilitate clinician-model interactions, enhancing decision-making. Following training, *MeDT* can generate tailored treatment recommendations by conditioning on the desired positive outcome (survival) and user-specified short-term stability improvements. We carry out rigorous experiments on data from the MIMIC-III dataset and use off-policy evaluation to demonstrate that *MeDT* recommends interventions that outperform or are competitive with existing offline reinforcement learning methods while enabling a more interpretable, personalized and clinician-directed approach.

## 1 Introduction

Sepsis is a fatal medical condition caused by the body's extreme response to an infection. Due to the rapid progression of this disease, clinicians often face challenges in choosing optimal medication dosages. Hence, there is significant interest in developing clinical decision support systems that can help healthcare professionals in making more informed decisions (Sutton et al., 2020). In the medical field, many tasks involve sequential decision-making, such as evaluating a patient's evolving condition in the intensive care unit (ICU) to make informed medical interventions. This is where reinforcement learning (RL) comes in as a promising solution for developing policies that recommend optimal treatment strategies for septic patients (Raghu et al., 2017; Komorowski et al., 2018; Killian et al., 2020; Saria, 2018; Huang et al., 2022).

These tools are intended to bolster and assist healthcare workers rather than replace them (Gottesman et al., 2018). Therefore, the reward function employed by these RL algorithms ideally necessitates clinician input to ensure that the policy generates decisions aligned with the domain expert's intentions (Gottesman et al., 2019). However, the majority of existing studies predominantly depend on binary reward functions, signifying the patient's mortality (Komorowski et al., 2018; Killian et al., 2020; Tang et al., 2022). In other words, the reward at each timestep in the patient's history remains zero until the final interval of the episode. This design leaves no room for clinician input to modulate the policy toward the achievement of desirable tasks, such as the stabilization of certain vital signs.

Existing works (Killian et al., 2020; Lu et al., 2020; Li et al., 2019) often rely on modeling the patient's medical history using recurrent neural networks (RNNs). These networks struggle with complex and long medical records due to vanishing or exploding gradients (Pascanu et al., 2013), leading to sub-optimal RL

policies (Parisotto et al., 2020). Sparse rewards also pose challenges in the learning of optimal policies since it can be difficult to identify a causal relationship between an action and a distant reward (Sutton & Barto, 2018). The sequential design of RNNs aggravates this problem. The low interpretability of model reasoning is another problem, given the high-stakes nature of clinical decision making. It is essential to address these challenges to create reliable decision support systems and improve clinical uptake of machine learning solutions. Transformers (Vaswani et al., 2017) are shown to effectively model long sequences, which enables learning of better representations for treatment histories of patients, potentially yielding more informed predictions.

In this paper, we propose the *medical decision transformer (MeDT)*, an offline RL framework where treatment dosage recommendation for sepsis is framed as a sequence modeling problem. *MeDT*, as shown in Figure 1, is based on the decision transformer (DT) architecture (Chen et al., 2021). It recommends optimal treatment dosages by autoregressively modeling a patient's state while conditioning on hindsight returns. To provide the policy with more informative and goal-directed input, we also condition *MeDT* on one-step look-ahead patient acuity scores (Le Gall et al., 1993) at every time-step. This enhances the potential for more granular conditioning while facilitating the interaction of domain experts with the model.

Below we summarize the main contributions of this work:

- We propose *MeDT*, a transformer-based policy network that models the full context of a patient's clinical history and recommends medication dosages.

- We develop a framework to enable clinicians to guide the generation of treatment decisions by specifying short-term target improvements in patient stability, which addresses the sparse reward issue.

- We demonstrate that *MeDT* outperforms or is competitive with popularly used offline RL baselines over multiple methods of off-policy evaluation (OPE) such as fitted Q-evaluation (FQE), weighted doubly robust (WDR) and weighted importance sampling (WIS). Additionally, we leverage a transformer network, the *state predictor*, to serve as an approximate model to capture the evolution of a patient's clinical state in response to treatment. This model enables autoregressive inference of *MeDT* and also serves as an interpretable evaluation framework of models used for clinical dosage recommendation.

## 2 Related Work

### 2.1 RL for Sepsis Treatment

The use of RL in sepsis treatment aims to deliver personalized, real-time decision support. It involves modeling optimal strategies for the administration of treatments, such as vasopressors (VPs) and intravenous fluids (IVs), based on patient data and expert advice. This problem poses a considerable challenge due to the potential for long-term effects associated with these treatments, such as the accumulation of interstitial fluid and subsequent organ dysfunction resulting from excessive fluid administration (Gottesman et al., 2018).

To address this issue, Komorowski et al. (2018) propose a value-iteration algorithm using discretized patient data from electronic health records (EHRs) for treatment action selection. Subsequent work uses Q-learning with continuous states and discrete actions and employs OPE to evaluate policies (Raghu et al., 2017). Huang et al. (2022) uses deep deterministic policy gradient (DDPG) with continuous states and actions to provide precise dosage recommendations. Other works explore model-based RL (Peng et al., 2018) and combined deep RL with kernel-based RL (Raghu et al., 2018) to further improve treatment recommendations for septic patients. Yet, several significant issues still need to be resolved, which currently impede the practical implementation of RL for the treatment of sepsis. Most of these studies assume that agents begin with a baseline reward of zero until the end of treatment. At the final time-step in a patient's history, a positive reward is given for survival and a negative reward otherwise. Since the manifestation of treatment outcomes (mortality) can occur with a delay of several days after decisions are made, it is challenging to identify effective treatment strategies. Shorter-term objectives, such as the stabilization of vital signs, are often

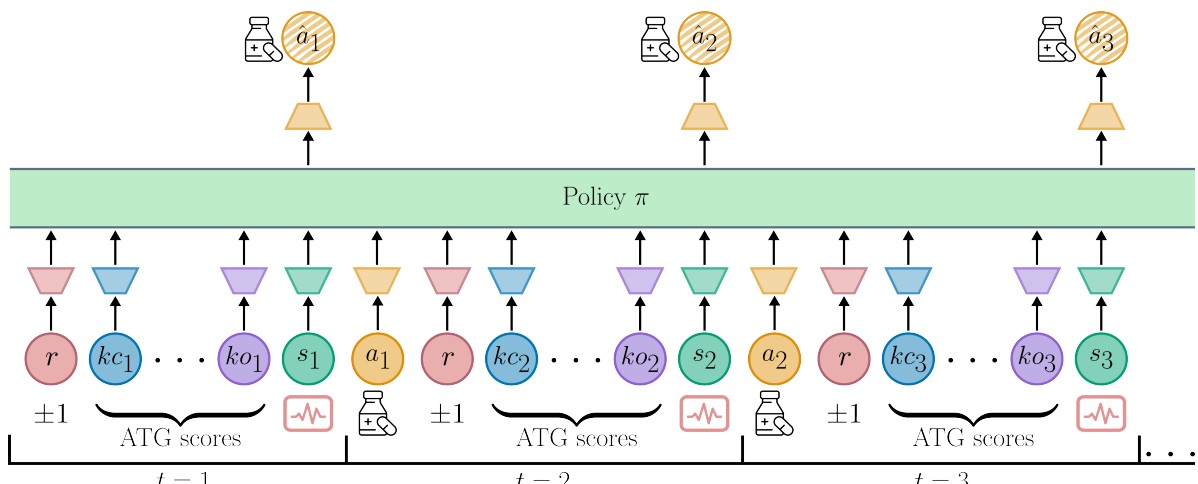

Figure 1: *MeDT* training: At each time-step $t$, the *MeDT* policy attends to the past treatment trajectory. This includes the desired treatment outcome $r$ (at inference time fixed to $+1$ indicating survival), desired next-step acuity scores $k_1, \ldots, k_t$ where $k_t = (kc_t, kr_t, kn_t, kl_t, kh_t, km_t, ko_t)$, patient states $s_1, \ldots, s_t$, administered drug doses $a_1, \ldots, a_{t-1}$, and outputs a dose prediction $\hat{a}_t$.

overlooked. Additionally, given the wealth of data being generated for each ICU patient, identifying the most relevant aspects in the treatment history may not be immediately apparent (Gottesman et al., 2018).

## 2.2 Transformer-based Policies

Another challenge in treatment modeling is introduced by the partial observability of the patient's state at each time-step. A single reading of vital signs provides incomplete information on the patient's well-being. RNNs address this issue by sequentially processing multiple time-steps of data, but face difficulties in capturing a patient's complete state history due to unstable gradients (Pascanu et al., 2013). This may result in incomplete information and, consequently, inaccurate decision-making (Yu et al., 2021). Recent research in RL (Parisotto et al., 2020; Parisotto & Salakhutdinov, 2021; Janner et al., 2021; Tao et al., 2022) is shifting towards attention-based networks (Niu et al., 2021) like transformers (Vaswani et al., 2017; Lin et al., 2022), which process information from past time-steps in parallel.

Transformers better capture long contexts and can be effectively trained in parallel (Lin et al., 2022). This addresses challenges posed by the sequential processing in RNNs (Wen et al., 2022). The self-attention mechanism in transformers is particularly beneficial, addressing issues related to sparse or distracting rewards. Self-attention, in short, first computes attention weights for information in each time-step by matching their corresponding *keys* and *queries*, which are learnable projections of input tokens. Afterwards, these weights are used to compute weighted sums of *values* corresponding to each time-step, potentially discovering dependencies between distant time-steps. DT (Chen et al., 2021) leverages these advantages for offline RL (Furuta et al., 2021; Xu et al., 2022; Meng et al., 2021), by conditioning a policy on the full history of states, actions and an observed or desired *reward-to-go*. Building on the DT architecture, we propose *MeDT*, which integrates additional conditioning via short-term goals for improvements in patient vital signs, yielding a framework for effective sepsis treatment recommendation.

## 2.3 Off-Policy Evaluation

OPE is a fundamental problem in RL concerned with estimating the expected return of a given decision policy using historical data obtained by different behavior policies (Uehara et al., 2022). Such an evaluation strategy is particularly useful in situations where interacting with the environment is costly, risky, or ethically challenging, like in healthcare (Sutton & Barto, 2018; Precup, 2000; Gottesman et al., 2020). However, OPE is inherently difficult because it necessitates counterfactual reasoning, i.e. unraveling what would have

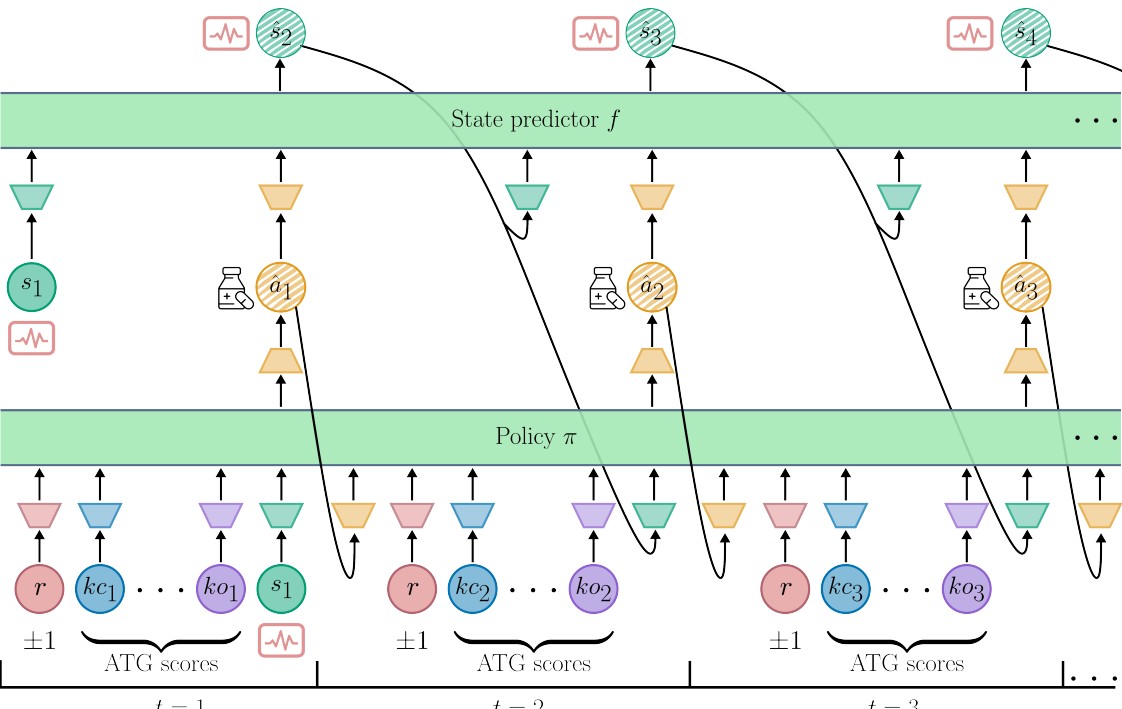

Figure 2: Autoregressive evaluation pipeline: At each time-step $t$, the pre-trained state predictor attends to past recommended doses $\hat{a}_1, \ldots, \hat{a}_t$, the initial patient state $s_1$ and predicted patient states $\hat{s}_2, \ldots, \hat{s}_t$, and outputs a prediction $\hat{s}_{t+1}$ of the patient state at time $t + 1$. Both dosage recommendations $\hat{a}_{t+1}$ and predicted states are fed back to *MeDT* to simulate treatment trajectories with multiple sequential decisions.

occurred if the agent had acted differently based on historical data. Nevertheless, while OPE may not necessarily help learn the optimal policy, it can help identify policies with lower suboptimality (Tang & Wiens, 2021).

A large subset of OPE methods are based on the concept of importance sampling (IS). IS uses validation data to assess the evaluation policy's value by adjusting the weight of each episode based on its relative likelihood (Pǎduraru et al., 2013; Voloshin et al., 2019). The WIS estimator is considered more stable than IS (Pǎduraru et al., 2013; Voloshin et al., 2019). On the other hand, we can directly estimate the evaluation policy's performance using the Q-function with FQE, rather than adjusting the weights of observed experiences like WIS (Le et al., 2019). FQE predicts the expected cumulative reward for taking a specific action in a given state. WDR is another OPE technique that combines two approaches for estimating policy value (Thomas & Brunskill, 2016; Jiang & Li, 2016). It leverages importance sampling, which adjusts the weight of past experiences based on their likelihood. Additionally, WDR incorporates value estimates at each step to improve accuracy and reduce overall variation in the learning process. Finally, approximate models (AMs) are another class of OPE that involves directly modeling the dynamics of the environment (Jiang & Li, 2016; Voloshin et al., 2019). This approximation, while not exact, may be sufficient to evaluate the policy's performance. Tang & Wiens (2021) empirically demonstrated that FQE provided the most accurate and stable estimations over varying data conditions. Given the difficult nature of evaluation in this problem setting, we utilize each of the four mentioned OPE methods to infer rigorous and robust policy evaluations.

## 2.4 Interpretability

The need for interpretability is more significant in safety-critical fields such as healthcare (Amann et al., 2020). Despite extensive research, the deployment of deep learning in healthcare has been met with resistance (Yin et al., 2021). This is primarily due to the black-box nature of these networks, resulting from their complexity and large number of parameters. Moreover, attaining interpretability in RL has been a major

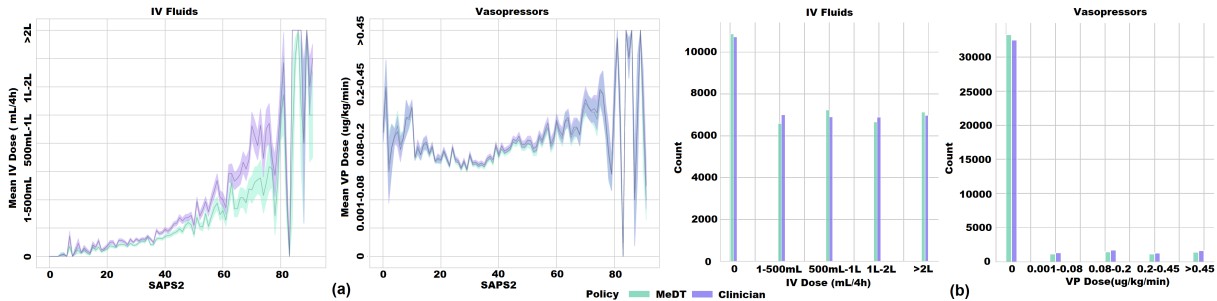

Figure 3: (a) Dosage recommended by *MeDT* and clinician policy for different SAPS2 scores. (b) Distribution of IV fluids and VPs given by the *MeDT* and clinician policies.

challenge hindering its deployment (Agarwal et al., 2023). Owing to their complexity, existing RL algorithms fall short of being fully interpretable (Glanois et al., 2021).

One simple method of attempting to understand the inner workings of transformers, is to visualize the computed attention weights. However, Serrano & Smith (2019) show that attention weights only produce noisy predictions of the relevance of each input token. Recent works delve into formulating methods that more representatively capture the relevance of input tokens. Abnar & Zuidema (2020) propose the rollout method, which considers paths over the pairwise attention graph while assuming that attention is computed linearly. However, this work is shown to assign importance to irrelevant tokens (Chefer et al., 2021b). Chefer et al. (2021b) introduce a method based on layer-wise relevance propagation, which is effective for encoder transformers. Chefer et al. (2021a) present a generic interpretability method that is compatible with every type of transformer architecture. The proposed method relies on the concept of information flow. It involves monitoring the mixing and evolution of attention to generate representative heatmaps, illustrating the importance assigned to input tokens in the model's decision. This method produces interpretations that are similarly or more accurate than prior methods while being simpler to implement. In this work, we utilize this method to generate interpretations of *MeDT* and visualize the relevance assigned in the input space to aid clinicians in understanding the rationale behind the model's decision-making.

## 3 Medical Decision Transformer (MeDT)

We frame our problem as a Markov decision process (MDP), comprising a tuple $(\mathcal{S}, \mathcal{A}, \mathcal{P}, \mathcal{R}, \mathcal{S}')$, where $\mathcal{S}$ denotes the set of possible patient states, $\mathcal{A}$ the set of possible dosage recommendations, $\mathcal{P}$ the state transition function, $\mathcal{R}$ the reward function and $\mathcal{S}'$ is the next patient state. While this framework is well-suited for learning policies via trial-and-error using RL methods, direct interaction with the environment can be risky in safety-critical applications like clinical decision making. To mitigate this risk, we use offline RL, a subcategory of RL that learns an optimal policy using a fixed dataset of collected trajectories each containing the selected actions, observed states and obtained rewards.

DT (Chen et al., 2021) uses transformers to model offline RL via an upside-down RL approach, where a policy has to select actions, that are likely to yield a specified future reward for a given past trajectory (Schmidhuber, 2019). Our proposed *MeDT* architecture follows a similar approach for learning policies.

The input tokens for the policy model encode past treatment decisions and patient states, as well as the desired returns-to-go (RTG). The output at each time-step is a distribution over possible actions. Specifically, we condition the model using RTG $r_t = \sum_{t'=t}^{T} R_{t'} = R_T$, which represents the singular positive or negative treatment outcome at the last time-step, similar to DT. In addition, we propose to condition *MeDT* on short-term goals, such as future patient acuity scores, or acuity-to-go (ATG). The acuity score provides an indication of the severity of the illness of the patient in the ICU based on the status of the patient's physiological systems and can be inferred from vital signs of the corresponding time-step. Higher acuity scores indicate a higher severity of illness. In this work, we opt to use the SAPS2 (Le Gall et al., 1993) acuity score as opposed to popular scores such as SOFA (Jones et al., 2009). This is because SAPS2 considers relatively

more physiological variables, which we believe will provide more informative conditioning, allowing flexible user interactions (Morkar et al., 2022). This formulation allows clinicians to input desired acuity scores for the next state, providing additional context for treatment selection. This leads to more information-dense conditioning, allowing clinicians to interact with the model and guide the policy's generation of treatment dosages.

To enable clinicians to provide more detailed inputs, we break down the SAPS2 score into constituent scores that correspond to specific organ systems (Le Gall et al., 1993). Following the definitions provided by Schlapbach et al. (2022), we define split scores $k = (kc, kr, kn, kl, kh, km, ko)$ to represent the status of the cardiovascular, respiratory, neurological, renal, hepatic, haematologic and other systems, respectively. A more detailed breakdown of these scores can be found in the Appendix in Table **??**.

In addition to the specification of the treatment outcome, our overall framework empowers domain experts to establish and select a scheme for interpretable short-term goal-conditioning, allowing clinicians to guide the model using their knowledge of the relation between intermediate goals, such as maintaining patients' vital signs within a specified range, and favorable long-term outcomes such as reduced mortality. This enhances the usability of the model for clinicians, enabling efficient interaction with the model for future dosage recommendations, considering the current state of the patient's condition. It is important to note that defining short-term goals presents a challenge, given the ongoing complexity in determining ideal targets for sepsis resuscitation (Simpson et al., 2017). Using these scores, the treatment progress over $T$ time steps forms a trajectory

$$\tau = ((r_1, k_1, s_1, a_1), (r_2, k_2, s_2, a_2), \ldots, (r_T, k_T, s_T, a_T)),\qquad(1)$$

where, for each time-step $t$, $r_t, k_t, s_t, a_t$ respectively correspond to the RTG, the ATG, the patient state, and the treatment decisions. We train our policy, a causal transformer network, to predict ground truth dosages that were administered by a clinician at each time-step given the treatment trajectory, ignoring future information and the prediction target via masking (Figure 1). *MeDT* aims to learn an optimal policy distribution

$$P_\pi(a_t|s_{\leq t}, r_{\leq t}, k_{\leq t}, a_{<t}),\qquad(2)$$

inspired by the model architecture used in Chen et al. (2021). We use an encoder with a linear layer and a normalization layer for each type of input (i.e. RTG, ATG, state, action) to project raw inputs into token embeddings. To capture the temporal dynamics of the patient's trajectory, we use learned position embeddings for each time-step, which are added to the token embeddings. Finally, the resultant embeddings are fed into a causal transformer, which autoregressively predicts the next action tokens.

## 3.1 Evaluation

In online RL, policies are typically assessed by having them interact with the environment. However, healthcare involves patients, and employing this evaluation method is unsafe. In this work, we evaluate the learned RL policy in an observational setting, where the treatment strategy is assessed based on historical data (Gottesman et al., 2018). Following the model-based OPE approach, we introduce an additional predictor network based on the causal transformer (Radford et al., 2018).

The predictor network, shown in Figure 2, acts as a stand-in for the simulator during inference. It is trained to learn a state-prediction model defined by the distribution

$$P_\theta(s_t|a_{<t}, s_{<t}),\qquad(3)$$

using a similar architecture as the policy network. This allows us to model how a patient's state changes in response to medical interventions. Rather than introducing a termination model, we use a fixed rollout length of $\mathcal{H}$. The estimated acuity scores can provide more clinically relevant estimates because they indicate how the stability of the physiological state of the patient may change given a treatment policy. While not exact, this approximation can prove adequate for generating reasonable estimates of a patient's physiological dynamics. This enables inferring estimates of patient acuity scores (SAPS2) from predicted states, which can then be used for policy evaluation. Furthermore, during inference, this model allows autoregressive

generation of a sequence of actions by predicting how the patient's state evolves as a result of those actions. Figure 2 and Algorithm 1 depict this rollout procedure.

---

**Algorithm 1** Evaluation Loop

---

1. **Input:** Initial patient state $s_0$

2. **Output:** Acuity score $g_1, \ldots, g_T$

3. Set target return $r_T = 1$

4. Initialize state $s_1 = s_0$, target return $r_{1:T} = r_T$ and action sequence $a_0 = \{\}$

5. **for** $t = 1, 2, \ldots, T$

6. Select desired Acuity To Go $k_t$

7. Select action $a_t = \text{MeDT}(r_{1:t}, k_{1:t}, s_{1:t}, a_{0:t-1})$

8. Append $a_t$ to the sequence of actions: $a_{1:t} = a_{0:t-1} + [a_t]$

9. Estimate new state: $s_{t+1} = \text{state\_estimator}(s_{1:t}, a_{1:t})$

10. Evaluate acuity score $g_{t+1}$ for state $s_{t+1}$

11. Append $s_t$ to the sequence of states: $s_{1:t} = s_{1:t-1} + [s_t]$

12. **end for**

13. **return** acuity score $g_1, \ldots, g_T$

---

Additionally, we use WIS (Pǎduraru et al., 2013; Voloshin et al., 2019), WDR (Jiang & Li, 2016; Voloshin et al., 2019) and FQE (Le et al., 2019) to rigorously evaluate the performance of policies.

WIS uses a behavior policy $\pi_b$ to evaluate a policy $\pi$ by re-weighting episodes according to their likelihood of occurrence (Pǎduraru et al., 2013; Voloshin et al., 2019). With the per-step importance ratio $\rho_t = \frac{\pi(a_t|s_t)}{\pi_b(a_t|s_t)}$ and cumulative importance ratio $\rho_{1:t} = \prod_{t'=1}^{t} \rho_{t'}$, WIS can be computed as

$$\frac{1}{N} \sum_{n=1}^{N} \frac{\rho_{1:T^{(n)}}^{(n)}}{w_{T^{(n)}}} \left( \sum_{t=1}^{T^{(n)}} \gamma^{t-1} r_t^{(n)} \right), \tag{4}$$

where $N$ is the total number of episodes, $T^{(n)}$ is the total number of time-steps for episode $n$, $\gamma$ is the discount factor and the average cumulative importance ratio $w_t = \frac{1}{N} \sum_{n=1}^{M} \rho_{1:t}^{(n)}$.

FQE is a value-based temporal difference algorithm that utilizes the Bellman equation to compute boot-strapped target transitions from collected trajectories and then uses function approximation to compute the $Q$ value of policy $\pi$. This can be formalized as

$$\frac{1}{N} \sum_{n=1}^{N} \sum_{a \in \mathcal{A}} \pi(a \mid s_1^{(n)}) \widehat{Q}_{\text{FQE}}^{\pi}(s_1^{(n)}, a), \tag{5}$$

where $\widehat{Q}_{\text{FQE}}$ is the estimated $Q$ function of $\pi$.

WDR utilizes both value estimators from FQE as well as importance sampling from WIS in order to reduce the overall variance of estimations (Jiang & Li, 2016; Thomas & Brunskill, 2016). The WDR estimator is defined as follows:

$$\frac{1}{N} \sum_{n=1}^{N} \sum_{t=1}^{T^{(n)}} \left[ \frac{\rho_{1:t}^{(n)}}{w_t} \gamma^{t-1} r_t^{(n)} - \left( \frac{\rho_{1:t}^{(n)}}{w_t} \widehat{Q}^{\pi} \left( s_t^{(n)}, a_t^{(n)} \right) - \frac{\rho_{1:t-1}^{(n)}}{w_{t-1}} \widehat{V}^{\pi} \left( s_t^{(n)} \right) \right) \right], \tag{6}$$

Table 1: Estimated final patient acuity scores (averaged over 2,898 patients) for BCQ: batch constrained Q-learning, NFQI: Neural Fitted Q-Learning, DDQN: Double Deep Q-Learning, CQL: Conservative Q-Learning, BC: behaviour cloning, DT: decision transformer and *MeDT*: medical decision transformer.

| Models | Overall ↓ | Low ↓ | Mid ↓ | High ↓ |
|--------|-----------|-------|-------|--------|
| BCQ | 42.10±0.03 | 41.78±0.07 | 42.38±0.03 | 41.59±0.15 |
| NFQI | 44.21±0.05 | 43.26±0.08 | 44.85±0.06 | 45.13±0.13 |
| DDQN | 43.47±0.04 | 43.08±0.07 | 43.64±0.05 | 43.29±0.11 |
| CQL | 40.42±0.03 | 40.18±0.07 | 40.59±0.04 | 41.03±0.11 |
| BC | 40.50±0.03 | 40.33±0.07 | 40.56±0.03 | 40.29±0.12 |
| DT | 40.38±0.03 | 40.16±0.06 | 40.49±0.03 | **40.06±0.12** |
| MeDT | **40.31±0.03** | **40.05±0.06** | **40.40±0.03** | 40.35±0.14 |

where $\widehat{Q}_{\text{FQE}}$ and $\widehat{V}_{\text{FQE}}$ is the estimated $Q$ and value function of policy $\pi$ respectively.

### 3.2 Interpretability

We utilize the transformer interpretability method introduced by Chefer et al. (2021a) which is based on the principle of information flow. We adapt this algorithm for the decoder transformer architecture of *MeDT* used in this work. This subsection outlines the mechanisms underlying the computation of relevance scores used to visualize interpretations.

Let $i$ refer to the input tokens of *MeDT*. $\mathbf{A}^{ii}$ represents the self-attention interactions between these tokens. Based on these interactions, we seek to compute the relevancy map $\mathbf{R}^{ii}$. Relevancy maps are constructed with a forward pass through the self-attention layers, where these layers attribute to aggregated relevance maps via the following propagation rules.

Given that each token is self-contained prior to attention operations, self-attention interactions are initialized with identity matrices. Thus, the relevancy maps are also initialized as identity matrices:

$$\mathbf{R}^{ii} = \mathbb{1}^{i \times i}. \tag{7}$$

The attention matrix $\mathbf{A}$ from each layer is used to update the relevance maps. The gradients $\nabla \mathbf{A}$ are used to average over the heads $h$ dimension of the attention map, to account for the differing importance assigned across the heads of the matrix (Voita et al., 2019). $\nabla \mathbf{A} := \frac{\partial y}{\partial \mathbf{A}}$, where $y$ refers to the output for which we wish to visualize relevance. The aggregated attention is then defined as:

$$\bar{\mathbf{A}} = \mathbb{E}_h \left( (\nabla \mathbf{A} \odot \mathbf{A})^+ \right), \tag{8}$$

where $\mathbb{E}_h$ is the mean over the $h$ dimension and $\odot$ is the Hadamard product. $^+$ denotes that the negative values are replaced by zero prior to computing the expectation.

At each attention layer, these aggregated attention scores are then used to calculate the aggregated relevancy scores as follows:

$$\mathbf{R}^{ss} = \mathbf{R}^{ss} + \bar{\mathbf{A}} \cdot \mathbf{R}^{ss}. \tag{9}$$

These relevancy scores can then be used to visualize the importance assigned across the input token space in the form of a heatmap. Since in transformer decoders, future tokens are masked, there is more attention toward initial tokens in the input sequence. Hence, to apply these methods to *MeDT*, we normalize based on the receptive field of attention.

## 4 Experiments

### 4.1 Experimental Settings

In this work, we train and evaluate the performance of *MeDT* on a cohort of septic patients. The cohort data is obtained from the medical information mart for intensive care (MIMIC-III) dataset (Johnson et al.,

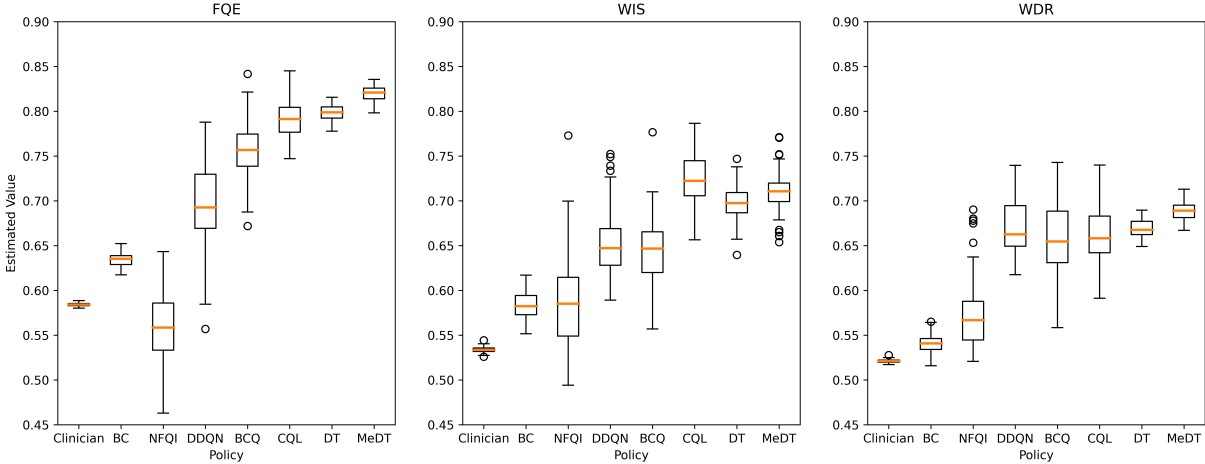

Figure 4: Box-plots of FQE, WIS and WDR off-policy evaluations for MeDT and baselines.

2016), which includes 19,633 patients, with a mortality rate of 9%. These patients were selected on fulfilling the sepsis-3 definition criteria (Singer et al., 2016). To pre-process the data, we follow the pipeline defined by Killian et al. (2020). We extract physiological measurements of patients recorded over 4-hour intervals and impute missing values using the K-nearest neighbor algorithm. Multiple observations within each 4-hour window are averaged.

The patient state consists of 5 demographic variables and 38 time-varying continuous variables such as lab measurements and vital signs. This work centers on the timing and optimal dosage of administering VP and IV fluids. The administration of each drug for patients is sampled at 4-hour intervals. We discretized the dosages for each drug into 5 bins, resulting in a combinatorial action space of 25 possible treatment administrations. Limiting our focus to IV fluids and vasopressors implies that these are the only treatments within our control; other interventions like antibiotics that the patient might receive are outside the scope of our consideration.

### 4.2 Baselines

We compare *MeDT* to batch constrained Q-learning (BCQ), neural fitted Q-learning (NFQI), double deep Q-learning (DDQN) and conservative Q-learning (CQL) algorithms, which are commonly used baselines in recent works related to offline reinforcement learning (Killian et al., 2020; Tang et al., 2022; Pace et al., 2023). Additionally, we train and evaluate DT and a transformer-based behaviour cloning (BC) algorithm. BC refers to a transformer that takes as input past states and actions, guided by cross-entropy loss on predicted actions, to directly imitate the behavior of the clinician's policy. DT builds on BC by conditioning on returns-to-go. The proposed *MeDT* differs from DT in that it also conditions on acuity-to-go at each time-step.

### 4.3 Training

The transformer policy is trained on mini-batches of fixed context length, which are randomly sampled from a dataset of offline patient trajectories. In our case, we chose a context length of 20, which is the longest patient trajectory in the dataset following pre-processing. For trajectories shorter than this length, we use zero padding to adjust them. During training, we use teacher-forcing, where the ground-truth sequence is provided as input to the model. At each time-step $t$, the ATG ($k_t$) is set to the actual acuity scores of the state at time-step $t+1$ in the sequence. The prediction head of the policy model, associated with the input token $s_t$, is trained to predict the corresponding discrete treatment action $a_t$ using a cross-entropy loss. The loss for a complete trajectory is averaged over time-steps. Additionally, the state estimator is separately trained to predict the patient's state following the treatment actions. The prediction head of the state predictor

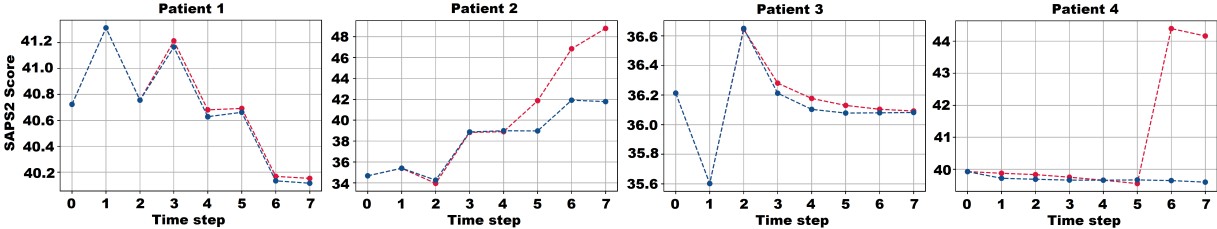

Figure 5: Visualization of 4 patient trajectories computed by the state predictor following treatment recommendation from DT (red) and *MeDT* (blue).

model, corresponding to the input token $a_t$, is trained to estimate the continuous state $s_{t+1}$ using a mean square error loss. The models are trained on NVIDIA V100 GPUs. We aggregate experimental results for each model into mean and standard error over five random seeds. Additional details on hyperparameter selection can be found in the Appendix in Section **??**.

## 4.4   Results and Analysis

We evaluate our proposed *MeDT* network in the autoregressive inference loop with the state predictor (Table 1). As elaborated in the Appendix in Section **??**, we use a naive heuristic to select ATG, and investigate whether the network conditioned on these prompts results in more stable patient outcomes. We compare our proposed approach to multiple baselines and run this loop over only 10 time-steps to avoid the accumulation of state-prediction errors resulting from the autoregressive nature of evaluation. We calculate the average and standard error of the SAPS2 scores of the states estimated by the predictor network for every patient in the test cohort. This cohort split comprises 2,945 patients. We also evaluate all policies over additional methods of OPE such as WIS, FQE and WDR.

### 4.4.1   Quantitative Analysis

From Table 1, we infer that the *MeDT* policy, which is conditioned on both positive RTGs and our chosen ATG heuristic, results in the most stable estimated patient states. The DT framework conditioned only with positive RTGs performs better than BC and other baselines. The learned policies are also evaluated for patients with different severity of sepsis (denoted as low, mid and high severity) based on the SAPS2 score of the initial state. Comparing the models, we observe that the *MeDT* policy results in more stable states for low and mid-severity patients, while DT performs best for high-severity patients. We hypothesize that, given there are far fewer data samples for patients in the higher severity bracket, *MeDT* was not able to learn an accurate mapping of patient states to actions given the additional ATG context. This suggests that *MeDT* requires more samples relative to DT to reach convergence.

We run an experiment to evaluate the sample efficiency of DT and *MeDT* in Figure **??** in the Appendix. We evaluate the performance of the policies when trained on 50%, 75% and 100% of the data from the train split. DT performs better when trained on the smallest 50% split, while *MeDT* is performant on the 75% and 100% splits. This supports the hypothesis that the additional conditioning used in *MeDT* has a negative impact on sample efficiency. It is worth noting that given the small size of the sepsis cohort from the MIMIC-III dataset, the 50%, 75% and 100% splits are all low training sample settings relative to standard sizes of training data used in RL. Nevertheless, this is an optimistic observation, given the potential for the exponential growth of data available from large-scale EHRs.

Figure 4 depicts the results of the FQE, WIS and WDR evaluations. The *MeDT* policy produces the highest estimated values on FQE and WDR while CQL performs best on WIS. It is worth noting that the *MeDT* and DT policies show noticeably less variance than the baselines, suggesting they are more robust models. These results indicate that the clinical dosage recommendations based on our proposed conditioning method may have had the intended treatment effects.

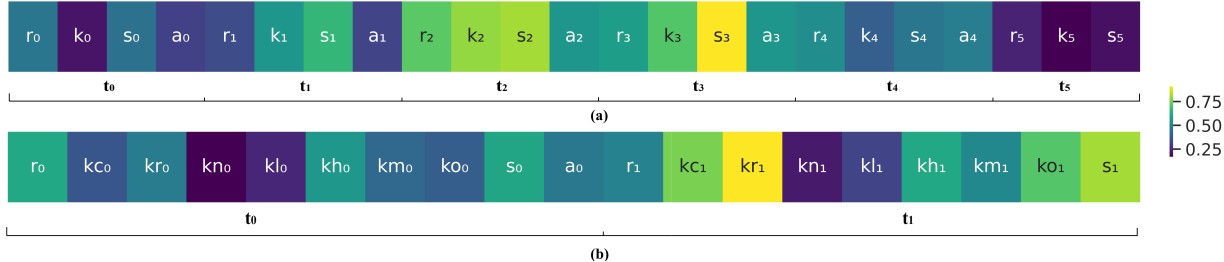

Figure 6: Relevancy maps depicting the importance assigned by the model to input tokens upon predicting action $a_5$ for a sample patient trajectory. Darker and lighter colors indicate lower and higher relevance scores, respectively. In (a) the mean of the relevancy assigned to the ATG components is taken to better visualize the relevance across time. Heatmap (b) depicts relevance across each ATG component to visualize the importance assigned to each conditioning.

### 4.4.2 Qualitative Analysis

We qualitatively evaluate the policy of *MeDT* against the clinician's policy. To ensure accurate analysis, we use ground-truth trajectories as input sequences instead of relying on autoregressive inference, which may lead to compounding errors. In Figure 3a, we conduct a comparative analysis of the mean dose of VPs and IVs recommended by the *MeDT* policy and the clinician policy, for patient states with varying SAPS2 scores. Figure 3b presents the dosage distribution of IVs and VPs recommended by both the *MeDT* and clinician policies.

Our results show that the *MeDT* policy generally aligns with the clinician's treatment strategy but recommends lower doses of IVs on average. Both policies exhibit a similar trend of increasing medication doses with worsening patient condition, for both VPs and IVs. Figure 3b reveals that the *MeDT* policy uses more zero dosage instances for both IVs and VPs, compared to the clinician policy. We hypothesize the signficant overlap between the *MeDT* and clinician policy is a byproduct of the imbalanced nature of the dataset, given that over 91% of patient trajectories in the dataset resulted in positive outcomes (survival). As a result, *MeDT* decides to imitate the clinician policy. Nevertheless, the alignment with the domain expert policy is ideal, especially in this high-stakes task where the algorithm relies solely on pre-existing static data for learning, as it is preferred to assess policies that only recommend subtle changes and closely resemble those of physicians as a precautionary measure (Gottesman et al., 2019).

Furthermore, previous studies have demonstrated a trend wherein lower dosages are recommended for patients with higher acuity scores (Raghu et al., 2017). This pattern can be linked to the common practice among clinicians of administering elevated dosages to individuals with high acuity scores, often associated with more severe medical conditions and, consequently, higher mortality rates. The challenge arises when algorithms lack data samples featuring high acuity scores coupled with minimal dosages. In such instances, these algorithms default to advocating lower medication doses. Figure 3 demonstrates that the *MeDT* policy diverges from prior research by refraining from recommending minimal dosages for patients with elevated acuity scores. This serves as an indicator of better generalization and sample efficient properties from *MeDT* given this negative behavior is not observed.

In Figure 5, we visualize the trajectories of multiple patients computed by the state predictor, following treatment actions recommended by both the DT and *MeDT* policies. The impact of ATG conditioning on patient health is evident, as *MeDT* leads to more stable trajectories, demonstrating the potential of our framework to generate targeted and improved treatment recommendations by considering both the hindsight returns and ATG at each time-step. In the Appendix in Figure **??**, we provide visualizations of some patient trajectories, where we observed that the *MeDT* policy produced the same or worse action policies relative to DT, with no discernible effect of ATG conditioning. We hypothesize that this may be due to limitations of the dataset, which may not sufficiently cover some regions of the joint space of vital signs, treatment decisions and outcomes, causing the model to be unable to discover some causal relations.

### 4.4.3 Interpretability

Currently, RL algorithms typically function as opaque systems (Gottesman et al., 2019). They take in data and generate a policy as output, but these policies are often challenging to interpret. This makes it difficult to pinpoint the specific data features influencing a suggested action. The lack of interpretability raises concerns, hindering experts from identifying errors and potentially slowing down adoption. Thus, clinicians may be hesitant to embrace recommendations that lack transparent clinical reasoning.

To improve interpretability and reliability of our *MeDT* model for users, we illustrate the relevance assigned by the transformer to input tokens for an example patient trajectory in Figure 6. The relevance across the ATG components are averaged in Figure 6a to better depict the relevance assigned over time, while Figure 6b visualizes the importance assigned to each ATG component. We observe that *MeDT* assigns relatively more importance to time-steps 2 and 3 for this patient sample. Figure Figure 6b shows that the model considers the conditioning for the Hepatic of higher relevance in its prediction.

This allows clinicians to monitor the specific points in time when the model assigns the highest importance, facilitating an assessment of its reasonableness. If the model differs from ground-truth clinician actions, an analysis may reveal which features carry the most weight in the decision-making shift. Additionally, if the model relies on clinically irrelevant features, it signals to clinicians that the recommendation may be unsound. This not only enhances understanding of the model's decision process but also invites future research into the reliability of deep RL decision-making from a clinical perspective.

## 5 Conclusion

In this work, we propose the *Medical Decision Transformer*, a novel reinforcement learning approach based on the transformer architecture. It models the full context of a patient's medical history to recommend effective sepsis treatment decisions. During training, our framework conditions the model not only on hindsight rewards but also on look-ahead patient acuity scores at each time-step. This enables clinicians to later interact with the model and guide its treatment recommendations by conditioning the model on short-term goals for patient stability. For autoregressive evaluation of our proposed approach, we present a separately trained state predictor that models a patient's clinical state evolution given a sequence of treatment decisions. Our experimental results demonstrate the potential of *MeDT* to bolster clinical decision support systems by providing clinicians with an interpretable and interactive intervention support system.

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
