# OpenReview forum: "Empowering Clinicians with Medical Decision Transformers: A Framework for Sepsis Treatment"
_TMLR — Rejected by TMLR_

### Review · Reviewer_zZqo · 2024-08-26

**Summary Of Contributions:**

The paper presents an approach called the Medical Decision Transformer (MeDT), which applies transformers to offline RL for sepsis treatment.

The MeDT model allows clinicians to interact with the model by setting short-term goals (like patient stability), addressing a common issue in medical AI—lack of transparency and control. This could may enhance the model's acceptance in clinical settings.

**Audience:**

Yes

**Broader Impact Concerns:**

I am not sure how the authors use the offline treatment data (i.e., MIMIC-III). The appendix section seems to be missing in this round of review.
This work may help on the future development of offline RL but it would be better adding a limitation section.

On the treatment data modeling, it is possible to calculate causal effect for the future studies in terms of broader impacts from causal RL communities.

**Claims And Evidence:**

Yes

**Requested Changes:**

In general, I enjoy reading this paper on how it position decision transformer for offline treatment modeling task. There are some directions would be improved but I think overall this paper is with good quality for TMLR submission.

- Suggestions

1. Since the method associated with medical decision making, there are less connection to treatment effect estimation [A] and causal discovery during the MDeT, it would be good to add some related references.

2. The references indexes have been done in Page 10 as showing `Section ??.`

3. The missing appendix.

4. On the interpretability score, it would be good to have some statistically refutation test for medical data modeling.

5. On the side of OffRL theories, there are less covered on the convergence and regret.

A. Training a resilient q-network against observational interference, AAAI 2022

B. Causal Reinforcement Learning: A Survey, TMLR 2023


- Minor

I would recommend to use less adjective to describe the experiment e.g., "rigorous."

**Strengths And Weaknesses:**

- Pros

1. The authors use a public set of off-policy evaluation methods, including fitted Q-evaluation (FQE), weighted doubly robust (WDR), and weighted importance sampling (WIS). This comprehensive evaluation supports the performance of their findings.


- Cons / Limitations

1. MeDT requires more samples to achieve convergence compared to other models like DT.

- Given the often limited availability of high-quality medical data, this could pose a challenge for the practical deployment of MeDT in settings with smaller datasets.

2. The paper mentions that the MeDT model underperforms in high-severity cases due to the lack of sufficient training data in this segment. This is one shortcoming since high-severity cases are where the need for accurate decision support is most crucial.

3. While the authors suggest that the model could be generalized to other conditions (i.e., OOD), the paper does not provide experimental evidence for this.

4. The handling of sparse rewards through conditioning on acuity scores is innovative, but it may introduce a bias if these scores are not well-calibrated across different patient populations. The potential impact of this sparse reward bias is not fully explored in the paper.

---

> ### Author Response · Authors · 2024-10-20
> **Response to Reviewer zZqo**
>
> We thank the reviewer for the constructive feedback. We are encouraged that the reviewer found the method evaluation to be comprehensive and that the overall paper to be of good quality. We address the reviewer's concerns below and will incorporate all feedback:
>
> **Missing Appendix**: We apologize for the confusion regarding the missing appendix, it was submitted as a supplementary material attachment. The missing reference was intended for a figure comparing the sample efficiency of MeDT and DT. The issue with the appendix links in the main paper has now been resolved, and the correct figure is now properly referenced in the document.
>
> **Reliability of Interpretability Scores**: To evaluate the reliability of the interpretability scores for medical data modeling, we conducted an experiment in which the timesteps ranked as most important by the interpretability method were iteratively masked. We then compared the impact on model performance to that of randomly masking timesteps. As illustrated in the figure linked below, masking the most important timesteps identified by the interpretability method led to a more significant degradation in model performance than random masking. The y-axis reflects patient outcomes, with higher values indicating worse patient conditions. This result suggests that the interpretability method is effective in reliably identifying critical timesteps that are crucial to the model's performance in medical data modeling.
>
> Figure: https://anonymous.4open.science/r/MeDT_rebuttal-0EAF/Interpretability_reliability.png
>
> **Impact of Conditioning on Subpopulation Bias**: To investigate for potential biases introduced by the proposed conditioning, we carry out an ablation study comparing behavior cloning (BC) with no conditioning, decision transformer (DT) with sparse return, and MeDT with both sparse return and dense ATG conditioning. This study focuses on model performance across subpopulations, particularly the gender attribute. The training set consists of 55% male and 45% female samples. This allows us to understand better how the conditioning influences performance across different groups. The results indicate MeDT shows a performance difference of only 0.09 between the two subgroups, which is very close to the values observed in BC (0.08) and DT (0.12) baselines. This suggests that the conditioning used in MeDT does not introduce a noticeable bias or performance disparity between male and female groups.
>
> |  | All | Male | Female | Difference |
> |-----------|-----------|-----------|-----------|-----------|
> | BC | 40.50±0.03 | 40.47±0.03 | 40.55±0.03 | 0.08 |
> | DT | 40.38±0.03 | 40.34±0.03 | 40.44±0.03 | 0.12 |
> | MeDT | 40.31±0.03 | 40.28±0.03 | 40.37±0.03 | 0.09 |
>
> **Limitations Section**: The limitations section was included in the Appendix of the supplementary materials. Based on the reviewer's feedback, we have updated this section to explicitly address certain drawbacks of our proposed method, as highlighted during the review. The ‘Limitations’ section of this work is as follows:
>
> "The MIMIC-III dataset has some limitations, as it only represents a specific geographic area, which could result in an over-representation of certain patient populations and an under-representation of others. Consequently, using the state predictor for evaluation may introduce biases inherent in the dataset on which it was trained. To mitigate these potential biases, we will investigate causal representation learning and pre-training techniques that enhance model robustness. Despite these limitations, MeDT provides a general framework to harness the vast amount of data found in large-scale EHRs from different modalities. Using the proposed framework, researchers can explore the scalability of the transformer architecture to develop systems for effective treatment recommendation for other medical conditions in the future."
>
> **Upside Down RL Theory on Convergence**: To address this, we have added the following text outlining the training objective of the MeDT policy:
>
> "Given the trajectories of conditioning, states and actions, we follow the formalism of upside-down RL and optimize a policy pi parameterized by theta trained with a supervised learning loss (cross-entropy), to map the conditioning (RTG and ATG) and state to the corresponding action:
> $$\arg \min _\theta \mathbb{E} _{s, a, r, k \sim D}[\mathcal{L}(a _{t},  P _{\pi}(a _t \mid s _{\leq t}, r _{\leq t}, k _{\leq t}, a _{<t};\theta))]$$
> The agent can attain a specific return and acuity-to-go by selecting actions from its stochastic policy that is conditioned on the desired hindsight returns and acuity scores."

---

> > ### Author Response · Authors · 2024-10-20
> >
> > **Generalization to Other Conditions**: Our goal was to highlight the flexibility of the conditioning input format, rather than suggest that the model is inherently designed to generalize across various medical conditions. The application of MeDT to other medical conditions is an area that requires further exploration and remains an open topic for future research. We will revise the text to clarify this point.
> >
> > **Performance of MeDT on High Severity**: We acknowledge the limitation that MeDT's performance declines in high-severity patients where the number of samples are relatively low. However, it is worth highlighting that MeDT avoids some of the undesirable behaviors observed in previous works towards high-severity cases. Prior studies (e.g., Raghu et al., 2017) have demonstrated that models often tend to recommend lower dosages for patients with higher acuity scores. This trend is largely influenced by clinical practices, where patients with more severe conditions typically receive higher dosages, which are also associated with higher mortality rates. The challenge for policies arises when there is insufficient data representing patients with both high-severity states and low-dosage treatments. In such cases, the policies struggle to learn from these sparse examples and default to recommending lower dosages. However, as shown in Figure 3, MeDT breaks away from this pattern by avoiding consistent recommendation of minimal dosages for high-acuity patients. This suggests that MeDT is able to generalize better in these scenarios, leading to more appropriate dosage recommendations for severe cases despite limited data.
> >
> > **Missing References**: We agree that treatment effect estimation and causal discovery are essential aspects of medical decision-making and should be discussed in the context of MeDT. To address this, we now add the following paragraph in the related works section on relevant works that explore treatment effect estimation and causal discovery in medical data modeling:
> >
> > "Causal machine learning has emerged as a crucial area of research in medical modeling, particularly in understanding and addressing the cause-and-effect relationships within medical data (Peters et al., 2017; Pearl, 2019). Unlike traditional predictive models, which primarily focus on associations, causal machine learning enables more informed decision-making by identifying how interventions directly influence outcomes. This is especially relevant in clinical settings, where understanding the causal impact of treatments on patient outcomes is critical. In the context of sepsis treatment, incorporating causal inference methods into reinforcement learning frameworks can improve model reliability by explicitly accounting for confounding factors and biases present in the data (Schulam and Saria, 2017). Recent advances have explored integrating causal structures with reinforcement learning, enhancing the robustness of policy recommendations in dynamic environments like healthcare (Lu et al., 2022). By leveraging causal inference techniques, models like MeDT can offer more interpretable and clinically relevant treatment strategies, ensuring that the recommended actions are both effective and aligned with the underlying causal mechanisms in patient care."
> >
> > **Sample Efficiency**: We acknowledge that MeDT underperforms compared to DT when trained on 50% of the original MIMIC-III dataset. This could be due to the increased complexity introduced by the ATG conditioning, which likely requires more training samples for the model to effectively learn the relationship between the additional conditioning, patient state, and actions. However, it is worth noting that even with 100% of the training set—around 19,000 patient trajectories—this still represents a relatively small dataset compared to traditional offline reinforcement learning benchmarks. Additionally, the growth of electronic healthcare records is accelerating, and given that the performance of MeDT is observed to scale better when presented with more samples, we anticipate that MeDT's performance gap to the baselines will further improve when applied to these larger datasets.

---

### Review · Reviewer_WncT · 2024-09-18

**Summary Of Contributions:**

The paper introduces the Medical decision transformer (MDT), which uses offline reinforcement learning for sepsis treatment recommendation. The main contributions stem from applying the transformer architecture to this problem, and introducing a more interpretable representation.

**Audience:**

No

**Claims And Evidence:**

Yes

**Requested Changes:**

Proposed changes (would simply strengthen the work):
1. Include results from the most important appendices
2. Highlight the novelty of this work

**Strengths And Weaknesses:**

Strengths:
The paper is well-written, and has useful implications for the healthcare domain
Weaknesses:
The paper lacks novelty, seems like a direct application of transformer models in this domain. The results presented in Table 1 do not seem significantly better than related works. Additionally the paper is lacking several references to tables in the appendix where more information is provided.

---

> ### Author Response · Authors · 2024-10-21
>
> We appreciate the reviewer's feedback and are pleased that they found the paper well-written with valuable clinical implications. Below, we address the specific concerns raised:
>
> **Appendix Links**: We apologize for the confusion caused by the broken links to the supplementary material. We've resolved the issue with the appendix links in the main paper, and the correct figures and sections are now properly referenced.
>
> **Novelty**: Our proposed method aims to provide a framework for medical dosage recommendation in sepsis treatment, making the following contributions: introducing a dense reward formulation to overcome credit assignment problems from sparse rewards; enabling domain expert inputs for fine-grained control of policy via acuity scores; and enhancing interpretability in clinical use.
>
> *Dense Rewards*: Existing works in RL for sepsis treatment predominantly rely on binary reward functions that signify patient mortality (Komorowski et al., 2018; Killian et al., 2020; Tang et al., 2022). Sparse rewards can pose credit assignment issues, making it challenging to determine which actions led to specific patient outcomes. We introduce a dense reward function based on acuity scores, providing more informative signals for the policy to generate actions. We conducted experiments over multiple offline RL baselines and evaluated the policies with a state predictor and several off-policy evaluation methods. The results demonstrate that MeDT consistently outperforms other models, including DT and other popular offline RL baselines.
>
> *Clinician Interactivity*: The proposed method enables clinicians to provide domain-relevant conditioning to the model at each time step, offering control over the policy's action generation. Clinicians can specify desired acuity scores for the next state, guiding the policy in treatment selection. By conditioning the model on one-step look-ahead patient acuity scores, we enhance the potential for precise control, allowing clinicians to input specific short-term goals. This approach lets clinicians leverage their knowledge to guide the model, aligning treatment recommendations with clinical intentions and patient-specific considerations.
>
> *Interpretability*: Current RL algorithms often lack full interpretability due to their complexity (Glanois et al., 2021). To address this, we provide interpretability relevance maps that visualize the importance assigned to each timestep in the trajectory input. By highlighting these timesteps, clinicians can more easily identify key moments in a patient’s medical history that might require further examination. We've also incorporated variable-level relevance analysis (Liu et al., 2022) to highlight specific patient state variables, such as vital signs and lab tests, that the model prioritizes during decision-making.
>
> Figure: https://anonymous.4open.science/r/MeDT_rebuttal-0EAF/interpretability_rebuttal.png
>
> We validated the reliability of the interpretability maps through an experiment where we iteratively masked the timesteps identified as most important by the interpretability method and compared the effect on downstream performance to random masking. As shown in the figure, masking the most important timesteps caused a more significant degradation in performance compared to random masking.
>
> Figure: https://anonymous.4open.science/r/MeDT_rebuttal-0EAF/Interpretability_reliability.png

---

### Review · Reviewer_h9Fd · 2024-10-07

**Summary Of Contributions:**

The paper presents a study on the use of decision transformers in the context of sepsis treatment, a common problem for evaluating the use of reinforcement learning in clinical scenarios.
Compared to previous approaches, one of the main novelties is the use of patient acuity scores.
The paper provides a thorough review and argumentation for the benefits of decision transformers in this context, and presents an empirical evaluation of both the qualitative and quantitative benefits of the proposed method.

**Audience:**

Yes

**Claims And Evidence:**

No

**Requested Changes:**

Minor issues: abbreviations are not explained / hyperlink package seems to be broken.

**Strengths And Weaknesses:**

Caveat: I am not an expert in the medical domain studied in this paper, but generally proficient in the field of RL.

The paper presents the necessity of improved methods for sepsis treatment in a relatively thorough background section, that nicely elaborates on the shortcomings of previous methods. The idea behind MeDT is intuitive to me and in general, the paper is relatively cleanly written and easy to understand.

However, while the paper provides a long list of purported benefits of the use of decision transformers in sepsis prediction, I found the claims of the paper only weakly supported.

As an example, it claims that one of the challenges of using RL in medical treatment is that of partial observability in the introduction of transformer based policies (2.2). Transformers do not necessarily change anything about the obervability of the environment, unless the assumption is made that a finite, fixed length history of the trajectory is sufficient to reconstruct the whole state of the patient. While transformers can be more effective at modelling sequences, it is unclear to me that they are necessarily better in this use-case. The paper does not provide an ablation for different methods such as BC etc. using different ways to deal with the past context, making it hard to ascertain whether the purported benefits arise from the use of a transformer, or other components of the method.
The fact that RNN are a rather weak backbone for reconstructing patient state was also already mentioned in Killian et al 2020, where another architecture was shown to be much stronger. A proper comparison should take these findings into account.

While the paper presents an evaluation on the benefits of using transformers for interpretability, I am not sure how the presented figures support that claim. As a non-domain expert in medicine, I am unable to verify whether the importance weights a) reflect known medical patterns and b) whether they reflect accurate internal processes of the model. As the claim of higher interpretability is much repeated throughout the paper, I would strongly encourage the authors to present a clinical user study with experts, or at least some more discussion on the usefulness of the importance weights, on whether this model does indeed provide improved usability for domain experts.

The numerical results also lack both context for easy evaluation and do not seem to fully support the authors claims. The authors report standard error estimates on their estimated performance. Leaving aside the issue of normalcy of the random variable, at a standard $3 \sigma$ confidence level, DT and MeDT overlap for all metrics presented in table 1, meaning no strong conclusion about the performance of the method can be drawn with the data as presented. Similar issues appear e.g. in the appendix in the data ablation experiment where both methods seem to perform exactly the same, especially given the low number of seeds.

Finally, the paper sets up three goals: using (decision) transformers in the context of sepsis treatment, using acuity scores as a more granular input to the agent, and interpretability in the context of clinical use. The experiments do not address these goals cleanly and do not provide separate ablations and evaluations for each. I believe either expanding the paper, or focusing cleanly on one of the goals would greatly strengthen the paper.

---

> ### Author Response · Authors · 2024-10-20
>
> We thank the reviewer for the insightful comments and constructive feedback. We address the raised concerns below and will incorporate all feedback:
>
> **Improved Model Interpretability**: We have revised the interpretability relevance representation to focus on averaged relevancy scores per timestep, rather than individual tokens. Displaying relevance at the token level often results in noisy saliency maps, making it challenging for clinicians to interpret and extract meaningful insights. By emphasizing which timesteps the model focuses on, clinicians can more easily identify key moments in a patient’s medical history that may require closer examination for potential abnormalities. In addition to timestep-level relevancy, it is important to identify which specific state variables, such as vital signs and lab tests, the model is prioritizing when making decisions. To address this, we’ve implemented a variable-level relevance analysis (Liu et al. (2022)). This approach highlights the most relevant patient state variables influencing downstream predictions during training and validation.
>
> Specifically, for each patient state at timestep $t$, $x_{t}$​, we mask the $i_{th}$ covariate value of the patient state, $x_{t}^{i}$​, by replacing it with the mean value of $x^{i}$ across all samples in the dataset. The patient state at timestep $t$ with the masked $i_{th}$​ variable is represented as $x_{t}^{i}$​. The contribution of each variable at timestep $t$, $R_{x_{t}}$​​, is then computed by comparing the outputs of the masked and non-masked covariates for each timestep as follows:
>
> $$R _{x^{i} _{t}} = (L(y^{‘} _{x _{t}^{i}}, y) - L(y’,y)),$$
>
> $$R _{x _{t}} = Softmax(R _{x^{1} _{t}}, R _{x^{2} _{t}}, R _{x^{3} _{t}}, …, R _{x^{n} _{t}}),$$
>
> where $n$ is the total number of variables in the patient state. The relevance of each variable is determined by calculating the difference between the policy's loss when all variables are available and the loss when the model is provided with the masked patient state. Applying the softmax function to these differences yields the relevance scores across the variable space of the patient’s state, allowing us to highlight the most critical variables contributing to the model's decision-making process.
>
> Fig: https://anonymous.4open.science/r/MeDT_rebuttal-0EAF/interpretability_rebuttal.png
>
> **Transformers vs RNNs**: We thank the reviewer for highlighting this point. In response, we have conducted an additional ablation experiment comparing the performance of a transformer and RNN, both trained as behavior cloning policies. We investigate the effect of varying the available context on downstream performance. The performance was evaluated based on two criteria: the estimated stability of the patient as predicted by the state transition model, and the accuracy of the policies in replicating the ground truth clinician actions. The brackets represent the number of parameters for each respective model.
>
> Estimated SAPS2 scores:
> |  | Transformer (827k) | RNN (337k) |
> |-----------|-----------|-----------|
> | Full Context | **40.50** | 43.57 |
> | Zero Context | 44.81 | 44.86 |
>
>
> Accuracy:
> |  | Transformer (827k) | RNN (337k) |
> |-----------|-----------|-----------|
> | Full Context |	**94.56%** | 86.41% |
> | Zero Context |	78.77% | 73.27% |
>
>
> From the ablation experiment, we observe that when making predictions with zero context, the transformer and RNN policies result in similar performance. Upon providing the policies with full context of the trajectory, we observe a sharper improvement in performance with the transformer policy. This suggests that the transformer model is better at utilizing past information than the RNN model.
>
> Additionally, we compared the performance of the transformer and RNN when trained as state transition models, evaluating their mean squared error (MSE) on the held-out patient test corpus.
>
> State transition modeling:
> |  | Transformer (827k) | RNN (337k) |
> |-----------|-----------|-----------|
> | MSE |	**0.271** | 0.384 |

---

> > ### Author Response · Authors · 2024-10-21
> >
> > **Marginal Performance Gains**: We acknowledge that the performance gains over DT in our state predictor evaluation are not highly significant. However, we conducted further experiments across several offline RL baselines and evaluated our model using more methods of OPE for a robust assessment. The results demonstrate that MeDT consistently outperforms DT and other popular offline RL baselines, while also providing clinicians with additional control over the policy through our proposed conditioning, which is the main motivation of our work. Additionally, we introduce a dense reward function based on acuity scores, offering more informative signals for the policy to generate actions. This helps avoid credit assignment issues that make it challenging to determine which actions led to specific patient outcomes.

---

### Decision · Action_Editor_Jp1e · 2025-01-08

**Recommendation:** Reject

**Comment:**

This paper introduces the Medical Decision Transformer (MeDT), applying decision transformers to offline reinforcement learning for sepsis treatment recommendations. While the approach is relevant and intuitive, addressing key challenges like interpretability and personalization, the reviewers generally agree that the claims are not well supported. In particular, the empirical results show only marginal improvements over existing methods, often overlapping with baselines, and lack rigorous ablations to isolate the contributions of key components. Furthermore, claims of improved interpretability, a central focus, are inadequately supported by clinical validation or user studies, leaving their practical value in question. Claims of generalizability to other conditions remain speculative without experimental evidence. In its current form, the paper provides interesting directions but lacks the necessary depth and validation to support its claims, making it premature for publication.

**Audience:**

yes

**Claims And Evidence:**

no (see meta-review)